# Recovering Climate Data from Documentary Sources: A Study on the Climate in the South of Spain from 1792 to 1808

**Fernando S. Rodrigo** 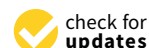

Department of Chemistry and Physics, University of Almería, 04120 Almeria, Spain; frodrigo@ual.es

**Abstract:** This work analyses new climate data on Southern Spain during the period 1792–1808. The data source is the periodical *Correo Mercantil de España y sus Indias* (Mercantile Mail of Spain and the Spanish Indies), which published weekly summaries of the weather conditions in Spain over this period. The study focuses on the southern provinces, providing 2788 new records, some of them corresponding to areas with no previously recorded data (Córdoba, Jaén). The analysis indicates the predominance of cold and dry winters, cold and wet springs, warm and dry summers, and variable autumn conditions, cold and humid in the western provinces while warm and dry in the eastern provinces. Some examples of these situations are discussed.

**Keywords:** historical climatology; southern Spain; the Dalton Minimum

## 1. Introduction

The period 1790–1830 is called the Dalton Minimum of solar activity [1]. During this period, there was also intense volcanic activity [2]. It is therefore a historical moment when natural radiative forcing could have significantly affected the global climate [3].

The Mediterranean Basin, and in particular the Iberian Peninsula (IP), have been recognized as areas particularly vulnerable to climate change [4]. Due to its geographical and latitudinal position at the western end of the Mediterranean Basin, the IP climate is governed by flows of Atlantic and Mediterranean origin, modulated by a varied topography with strong altitudinal gradients in a relatively small area [5]. As a result, the study of the IP climate during a period of natural climate change such as the Dalton Minimum is particularly interesting.

Although the objective of historical climatology is to reconstruct long and continuous data series which overlap with modern instrumental series, in order to achieve proper calibration and validation [6], the value of short series is now recognized [7]. These generate useful reconstructions of particular years (e.g., the "year without a summer" of 1816), contributing to a greater understanding of extreme events, and of processes such as the transition between different climatic periods, the climatic influence of volcanic eruptions, and intra-annual and interdecadal variability, as well as connecting climate data with their social impacts and responses.

The main sources of meteorological data for this period are the early meteorological data series (EMD) collected by individual efforts and at the initiatives of scientific societies interested in the relationships between climatic conditions, sailing, medicine and agriculture [8]. However, EMDs are heterogeneous, fragmentary, dispersed with incomplete information regarding the metadata, and only a few cities in Spain (Cádiz, Madrid, Barcelona) have long and continuous series of weather data [9]. Therefore, it is still necessary to find new data sources that report on weather conditions during this historical period.

Recently, a new source of historical climate data from Spain during the period 1792–1808 has been described [10]. This comes from the periodical *Correo Mercantil de España y sus Indias* (Mercantile Mail of Spain and the Spanish Indies; 1792–1808; CMEI hereinafter; Figure 1). The editors established a network of correspondents across the country to obtain information from most of their provinces. The periodical appeared twice a week, beginning on 1st October 1792 and ending on 30th June 1808, when the Napoleonic invasion interrupted its publication. The copies of this periodical are digitized and available on the National Library of Spain website [11], except for two large gaps from July to November 1798, and from July 1804 to July 1806 (we have not been able to locate these lost editions).

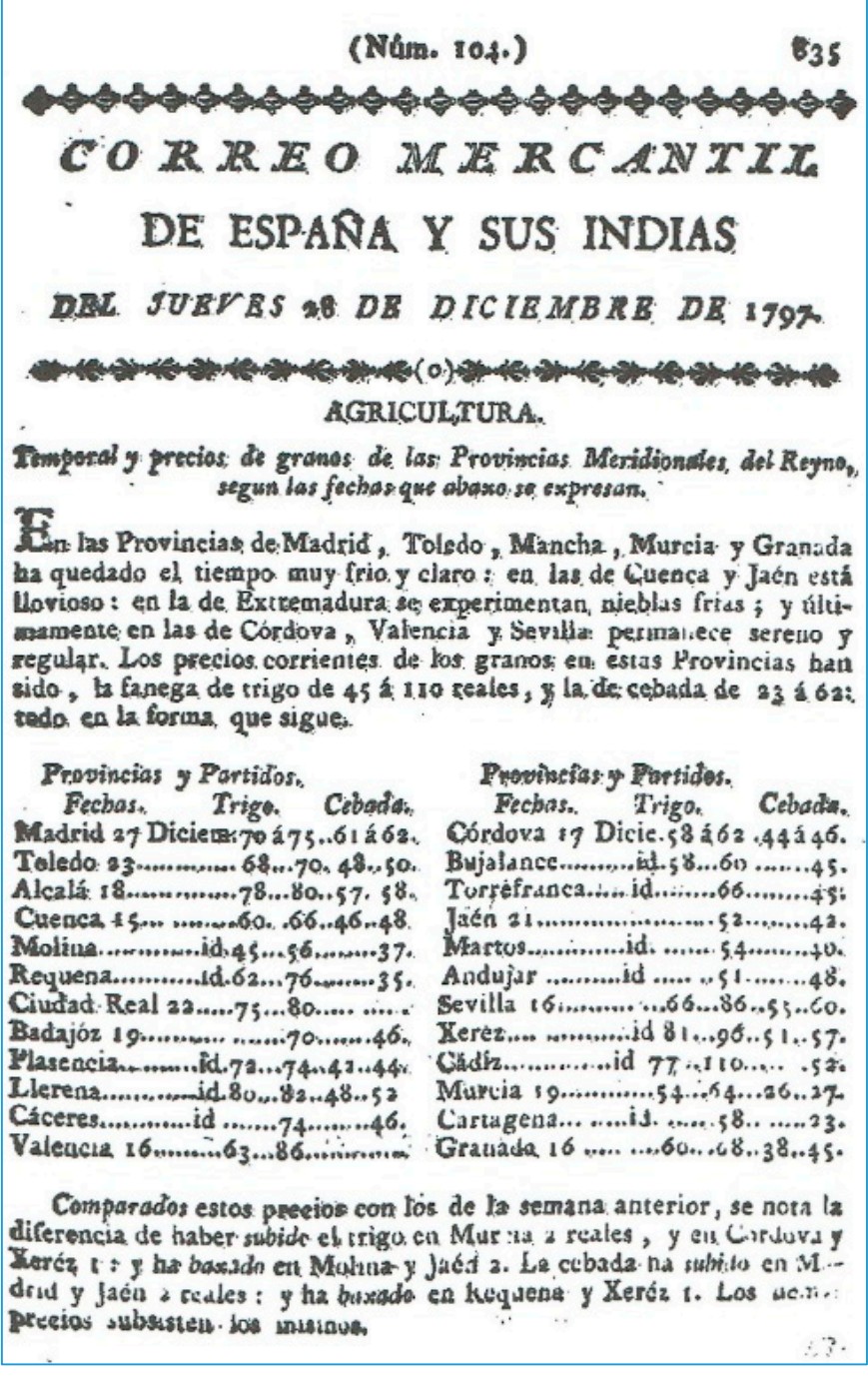

**Figure 1.** Front page of issue 104 (28/12/1797) of *Correo Mercantil de España y sus Indias* (CMEI).

All the newspaper's editions begin with an agricultural report, which describes weather conditions qualitatively and indicates the grain prices for each province. The interesting thing about this data source is that it provides weekly information on meteorological events in Spain during the middle years of the Dalton Minimum, providing information on areas that were lacking until now over a wide spatial range. In its first approximation, the work focused on the provinces of southern Spain (Sevilla, Córdoba, Jaén, Granada, and Murcia; Figure 2 [12]) to complete previous work dedicated to the EMD in this area [13]. In the present study, however, information from other IP areas will be used.

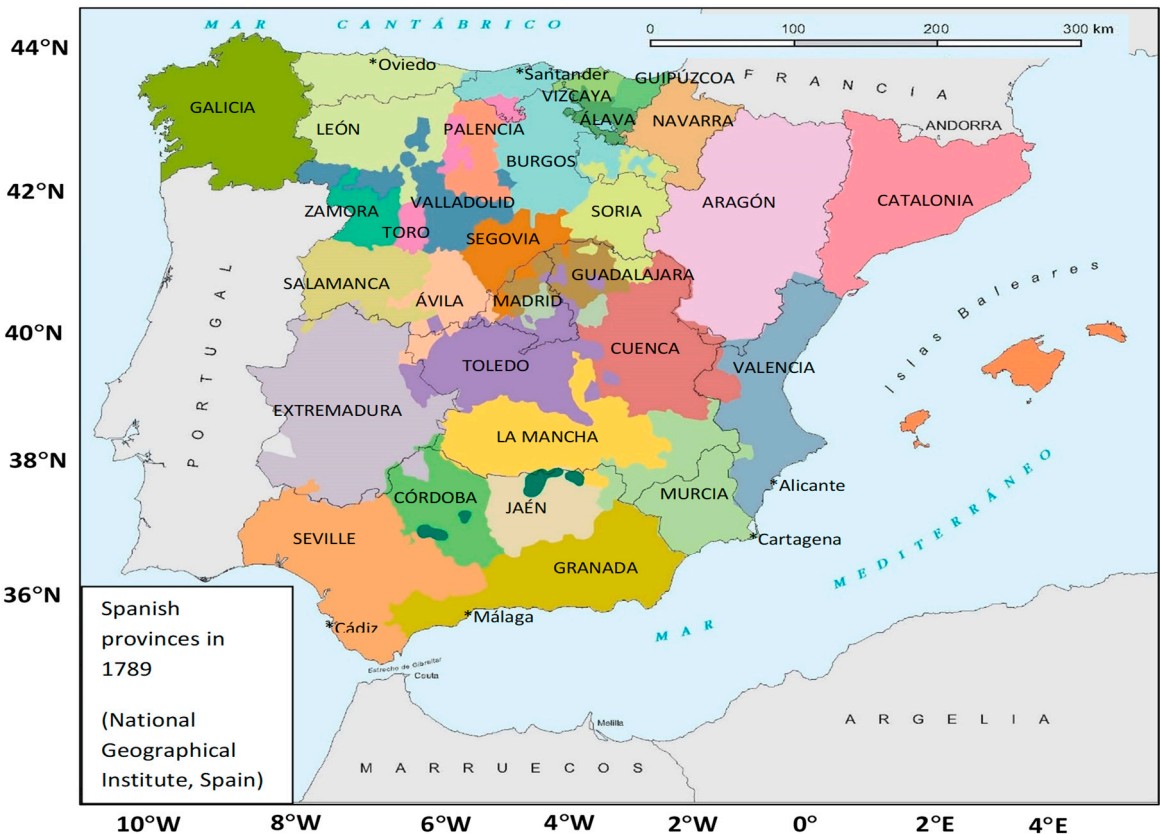

**Figure 2.** Spanish provinces in 1789 according to the Atlas Nacional de España [12].

In historical climatology, there are various methods to reconstruct climatic conditions by analysing documentary sources [14–19]. In our case, given the nature of the data source under study (a printed periodical), and its weekly resolution in which not only extreme phenomena are described, the most appropriate method is to analyse the weather descriptors used in the texts.

The outline of the article is as follows: Section 2 describes the weather information obtained and the working method while Section 3 describes the general climatic conditions obtained for the southern IP during the study period. These results are discussed in Section 4. Lastly, Section 5 summarizes some of the findings and research perspectives for the future.

## 2. Data and Methods

The concepts used for the data source in the descriptions of weather events are briefly described below; a more detailed explanation can be found in [10].

All the periodical's editions start with an agricultural report qualitatively describing the general weather conditions and indicating the grain prices. The information is classified by province, taking into account the territorial division at that time. The reports are a brief summary in which the editors give information for each province. Therefore, the information provided is basically a spatial "average", although sometimes differences are indicated between several localities within the same province.

The weekly resolution ensures that not only information on extreme phenomena was collected, but also descriptions of "normal" conditions during the study period. Thus, it is common to find comments such as "typical seasonal weather" or "according to the season". In these cases, at least, we can admit the absence of especially relevant extreme phenomena. Weather information was directly related to agricultural productivity, wheat in particular, from sowing in autumn to early summer harvest, although other crops (olives, fruit trees and livestock pastures etc.) also held the correspondents' attention. Different crops have different climatic requirements, as do the same crop depending on the phenological stage of plant growth. Thus, comments such as "fruit-friendly weather", or "appropriate for the plants", were not used in the study, given the difficulty of establishing the precise weather conditions that predominated in these cases.

Fortunately, many of the comments are direct descriptions of weather events, focusing primarily on the thermopluviometric regime, although other phenomena, such as the appearance of fog, winds, storms, cloudiness and atmospheric humidity are also recorded. Table 1 shows a summary of the main descriptors used in the texts. Each temperature-related record was indexed with the values $i_t =$ +2 (very warm), +1 (warm), 0 (mild), −1 (cold), −2 (very cold). In the case of rain, it was possible to establish a gradation from the absence of rain ($I_r = 0$), light and/or dispersed rain ($I_r = +1$), moderate rain ($I_r = +2$), and heavy rain ($I_r = +3$). Some descriptors are more diffuse and imprecise, such as "wet", "unsettled", "variable", or "fair" weather references. In these cases, only by comparing with other contemporary independent data sources can one clarify the true meaning of these concepts. These terms were not indexed to avoid possible circularity problems (circular reasoning) in contrasting this new data with the EMD series. We therefore preferred to use only those terms that unequivocally described the prevailing weather conditions.

Sometimes not all the provinces provided information, probably due to local correspondents not responding, problems with the mail or transportation, lost information or editorial errors. In addition, the information is not simultaneous between the different provinces, with gaps of several days between some provinces and others. This requires each province to be studied separately. In a first analysis, it was decided to look at the records corresponding to the southern provinces (Sevilla, Córdoba, Jaén, Granada, and Murcia) because EMD series exist for the same area over a similar period [13]. Hence, it is possible to compare the information from various independent sources and incorporate Córdoba and Jaén, regions with no data recorded until now, at least that we know of. The records were extracted from the data source, tabulated and indexed. As a first result, the database of meteorological observations in southern Spain was expanded [20]. The total number of new records is 2788, distributed as follows: 518 from Sevilla, 548 from Córdoba, 599 from Jaén, 598 from Granada and 565 from Murcia.

A monthly temperature (rain) index, $I_t$ ($I_r$), was defined as the average of the $i_t$ and $i_r$ indices assigned to each individual record. The monthly index was calculated only if the number of weekly records per month was ≥ 3 and considering that some weeks belonged to two consecutive months. In the time series obtained, the 13-month moving average was estimated to filter out intra-annual seasonal variations and obtain an initial estimate of each variable's temporal evolution.

The climate in Spain has a clear seasonality, with marked differences between winter (December to February) and summer (June to August), with spring (March to May) and autumn (September to November) as transitional seasons. Therefore, a separate study was done for each season of the year (in what follows, winters are identified by the year to which January to February correspond). For each year, the optimal number of seasonal records was set at ≥ 9, considering a sufficient number of monthly records to be ≥ 3 [21]. Figure 3 shows the methodology used to study the meteorological information, using the spring data from Córdoba as an example. Considering our criterion for the optimal number of seasonal records, the years 1795, 1802, 1803, 1805, and 1806 were eliminated from the analysis (Figure 3a). The next step was to take into account the percentage of records related to cold or warm weather ($i_t < 0$, $i_t > 0$, respectively; Figure 3b), and dry or rainy weather ($i_r = 0$, $i_r > 0$, respectively; Figure 3c). Our hypothesis here is that the general character of each season (warm/cold and/or dry/wet) is indicated by the highest percentage. Note that this classification indicates the

"average" conditions in each season of the year and in each province, not the possible extreme character of a particular season.

**Table 1.** Terms used in the CMEI to describe the meteorological events and the assigned indices ($i_t$ = temperature index; $i_r$ = precipitation index).

| Category | $i_t$ | $i_r$ | Descriptors (Spanish) | Descriptors (English) |
|---|---|---|---|---|
| Very warm | +2 | | Calores vehementes/fuertes/excesivos<br>Muy/excesivamente/bastante caluroso<br>Soles muy picantes/Ardiente | Vehement/strong/excessive heat.<br>Very/excessively/quite hot.<br>Hot/ burning sun. |
| Warm | +1 | | Caluroso/algo caluroso | Hot/rather hot. |
| Mild | 0 | | Templado/suave/bueno/fresco | Mild/soft/good/fresh |
| Cold | −1 | | Hielos/escarchas/nieve<br>Vientos fríos | Ice/Frost/snow.<br>Cold winds |
| Very cold | −2 | | Muy/excesivo/extraordinario/<br>bastante frío | Very/excessively/extraordinarily/quite cold |
| Clear | | 0 | Sereno/serenidad/seco/árido/sequedad<br>Soles/claro/despejado<br>Se necesita el agua | Calm/quiet/dry/arid/dryness<br>Sun/clear/ cloudless<br>In need of water |
| Cloudy (without rainfall) | | 0 | Nubes/nuboso/cubierto<br>Aparatos de lluvia/nieve | Clouds/cloudy/covered<br>Signs of rain/snow |
| Slight rainfalls | | 1 | Ha llovido algo/a veces/<br>en algunas partes<br>Ha llovido un poco<br>Ligeras lluvias/chubascos | It has rained/sometimes/in some places<br>It has rained a little<br>Light rains/showers |
| Moderate rain | | 2 | Lluvias/lluvioso<br>Ha llovido moderadamente/suficiente | Rains/raining<br>It has rained moderately/enough |
| Heavy Rain | | 3 | Ha llovido copiosamente/abundantemente/<br>excesivamente<br>Aguas excesivas/continuas/constantes | It has rained<br>copiously/abundantly/excessively<br>Excessive/continuous/constant rain |
| Fog | | | Nieblas | Fog |
| Wind (direction) | | | Vientos (solanos/levantes; nortes; sur; ponientes) | Wind (Easterly/levantine; northerly; southerly; westerly/ponente) |
| Other | | | Revuelto/vario/húmedo/regular/favorable | Unsettled/changeable/humid/fair/favourable |

It is interesting to consider the combined effect of the thermal and pluviometric (rainfall) regimes [22,23]. To do this, the differences between the percentages of wet and dry (warm and cold) weather records were calculated for each year and represented, as shown, in Figure 3d. In this figure, each point represents the combined character of a particular season. If the point is located in the first quadrant, it corresponds to predominantly warm–wet conditions, in the second quadrant to cold–wet, in the third to cold–dry, and in the fourth to warm–dry. If the point is located on the horizontal (vertical) axis, this indicates "average" rain (temperature) conditions. In our example, the springs of 1793, 1796, 1799, 1800, and 1801 can be considered cold–wet, while the spring of 1808 was warm–dry; those of 1797, 1798 and 1807 can be considered temperate and humid, and those of 1794 and 1804 temperate and dry. Therefore, during the study period, cold–wet springs predominated in Córdoba. This analysis was done for each season of the year and for each of the five provinces analysed (Figures S1 to S5, Supplementary Material).

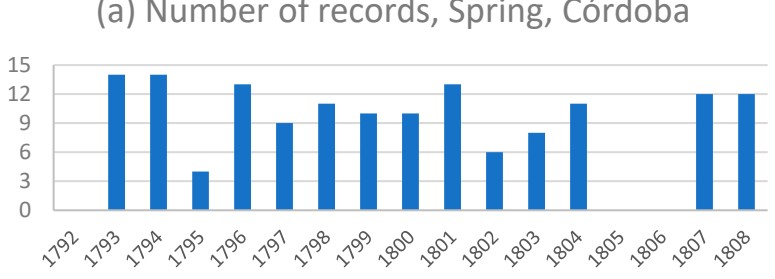

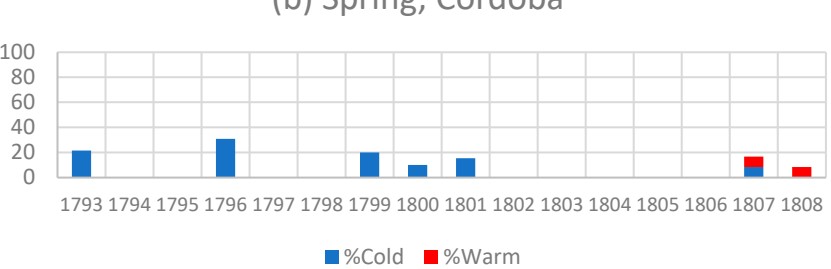

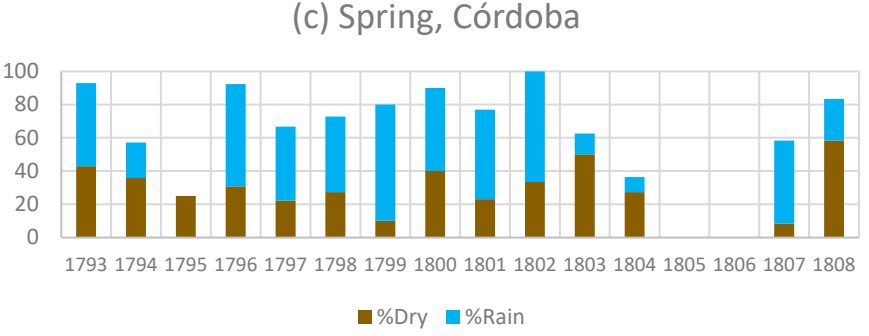

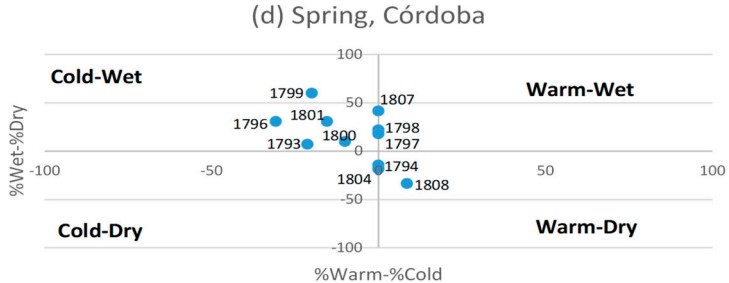

**Figure 3.** (**a**) Number of spring records in Córdoba from 1792 to 1808. (**b**) Percentage of records indicating cold (blue) and warm (red) conditions. (**c**) Percentage of records indicating dry and wet conditions. (**d**) Characterization of the combined temperature and rainfall conditions in Córdoba from 1792 to 1808. Years are indicated.

A feature of this data source is that it allows one to view the spatial coverage of the different events, expanding the information beyond the individual provinces. This is especially interesting in the case of extreme events and allows us to use those records that are not considered in the monthly indices. The information collected can be represented on a map describing the general weather conditions for the whole country on a weekly basis. In some cases, the analysis allows us to infer the underlying atmospheric dynamics. In the analysis, given the existence of time lags between the different provinces, the records are grouped, allowing lags of up to 1 day around the central day chosen. The following section provides some examples.

Information collected from documentary sources may be influenced by the subjectivity of the authors. The process of assigning ordinal indexes to the obtained records is an attempt to objectify the information collected as much as possible, although a certain degree of uncertainty always remains. Given the qualitative nature of the data, the study of its reliability (i.e., ensuring that what they describe actually occurred) is itself essentially qualitative. For this reason, we searched for information in other independent data sources that could confirm our data. One notable data source amongst these was reconstructions of the average pressure at sea level (SLP) on a monthly basis in the historical past [24]. These reconstructions allow us to interpret in a plausible way the dynamic atmospheric conditions that gave rise to the events in our maps. However, it is necessary to note that our maps represent events on a weekly timescale whereas SLP maps are monthly reconstructions, so perfect agreement cannot be expected.

There are other data sources that were explored in a previous work [13], which are also compiled in the "Early Meteorological Observations in Southern Spain" database (EMOSSv2 [20]). This is a set of qualitative descriptions and instrumental series from Cádiz, Sevilla, Málaga, Granada and Murcia. Comparing them is difficult because the length of the common periods for both datasets (the CMEI and EMOSSv2) is brief and the temporal resolution is different (weekly for the CMEI, daily and/or monthly for the EMOSSv2 series), as is the spatial resolution (the provincial average for the CMEI, the local average for EMOSSv2). Nonetheless, a comparison exercise was attempted. To do this, where possible, the monthly index $I_t$ was compared using a linear regression analysis with the monthly average temperature obtained from the EMOSSv2 instrumental data.

## 3. Climatic Conditions in Southern Spain from 1792 to 1808

### 3.1. Monthly Indices

Figure 4 shows the temporal series of the monthly indexes $I_t$ and $I_r$ for the five selected provinces. The 13-month moving average allows us to visualize the temporal evolution of the indices by filtering the intra-annual seasonal variations. Although gaps exist, especially around 1805, that prevent conclusive results from being established, the increase in the $I_t$ index from 1799 is evident. For the period as a whole, the average value of the $I_t$ index along with the standard error in the mean estimation is $-0.08 \pm 0.08$ in Sevilla, $-0.10 \pm 0.07$ in Córdoba, $+0.03 \pm 0.07$ in Jaén, $-0.05 \pm 0.08$ in Granada, and $+0.15 \pm 0.10$ in Murcia, suggesting the existence of a temperature gradient from west (colder conditions) to east (warmer conditions). The rainfall index $I_r$ shows a maximum around 1799–1800, and a clear decline in the eastern provinces, in Granada from 1799 and in Murcia practically from the beginning of the series. The average values for the $I_r$ index are $+1.12 \pm 0.08$ in Sevilla, $+0.98 \pm 0.07$ in Córdoba, $+0.91 \pm 0.07$ in Jaén, $+0.86 \pm 0.06$ in Granada, and $+0.72 \pm 0.06$ in Murcia. This result suggests a decrease in rainfall along the West–East axis, from those provinces most influenced by Atlantic flows (Sevilla, Córdoba) to the eastern end (Murcia), where mechanisms of Mediterranean origin predominate.

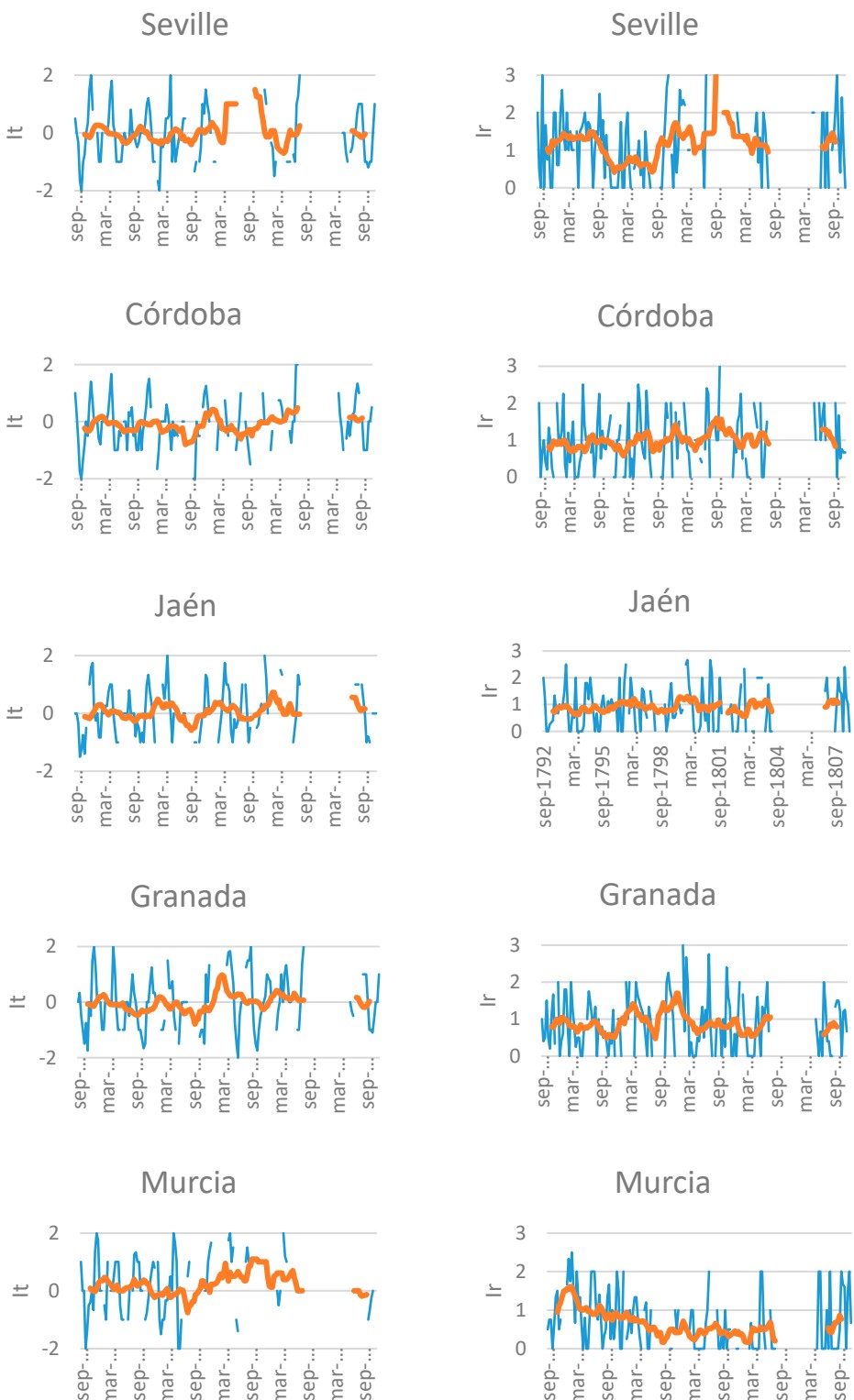

**Figure 4.** Thin line: time series of the monthly indices $I_t$ (left), and $I_r$ (right) for the five provinces studied. Thick line: 13-month moving average.

*3.2. Combined Variability Modes*

Figure 5 shows the distribution of the seasons of the year classified as warm–wet, warm–dry, cold–wet and cold–dry, for each of the seasons of the year and the five provinces selected. There was a marked seasonality (common to the five provinces), with cold winters, warm summers, and spring and

autumn as transitional seasons. The predominance of cold–dry conditions in winter (e.g., in 1793, 1796, 1798, 1801, and 1808), warm–dry conditions in summer (1796, 1803, 1807), and cold–wet conditions in spring (1796, 1799) is evident. The main differences appear in the autumn, indicating a certain transition from the west (wet conditions in Córdoba and Sevilla) to the east (warm-dry autumns in Murcia) for this season, with Jaén and Granada as transitional provinces. Thus, for example, the autumn of 1794 was cold–wet in Sevilla and cold–dry to the east, while the autumn of 1801 was cold–wet in Córdoba and warm–dry in Murcia.

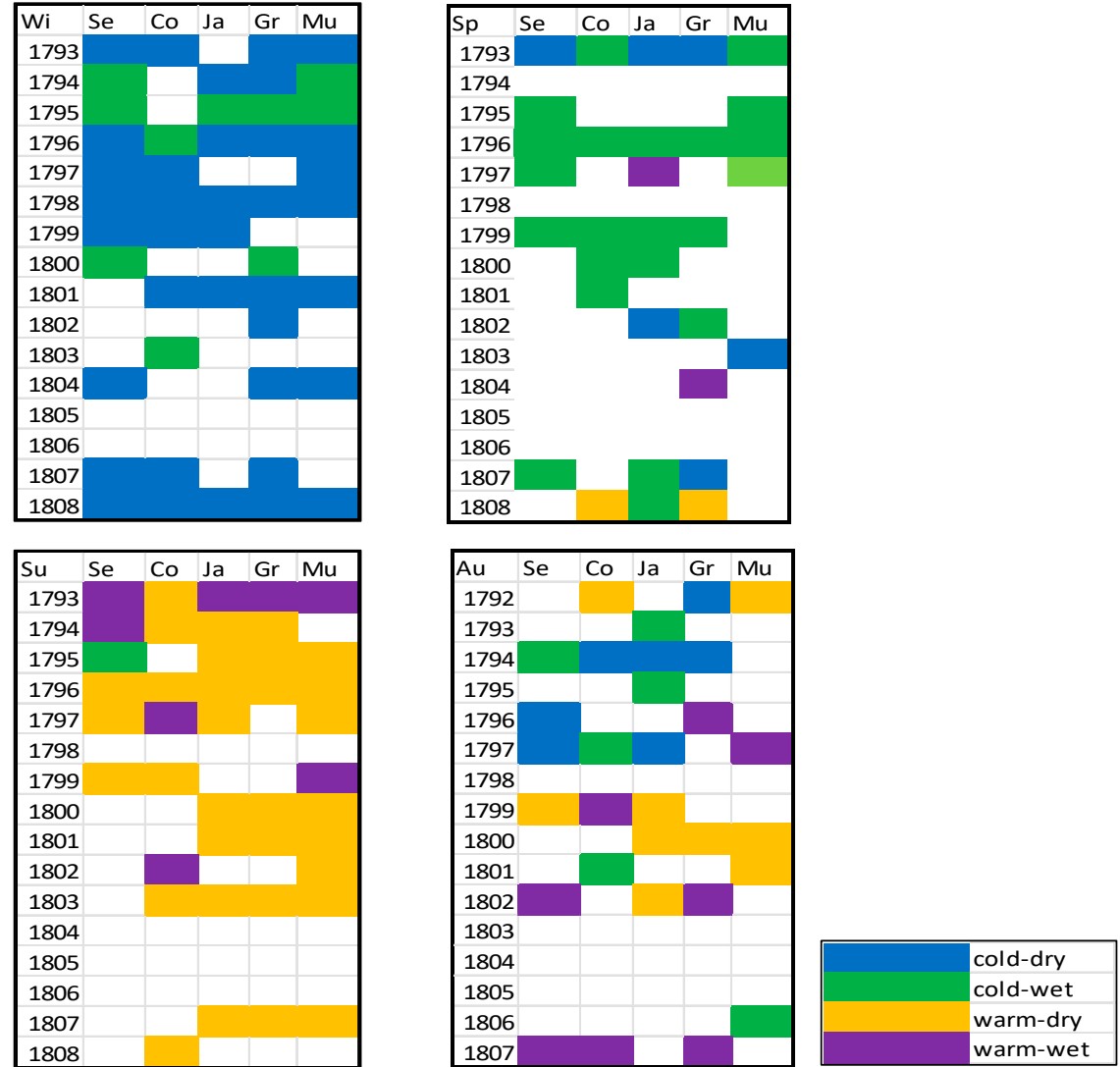

**Figure 5.** Combined temperature–rainfall character of the four seasons of the year (Wi = winter, Sp = spring, Su = summer, Au = autumn) and the five provinces studied (Se = Seville, Co = Córdoba, Ja = Jaén, Gr = Granada, Mu = Murcia).

*3.3. Examples*

During the period 1792–1808, there was a predominance of cold and dry winters in the study area. Figure 6a shows an example of this. The records dated between January 18th and 20th, 1799 were used to produce this map (Table S1 of the Supplementary Material gives comments for each province). Given the weekly nature of the records, these conditions correspond on average to the period from January 11th to 19th of that year. One can see that, in all of the provinces with information, there are references to cold and sunny weather, including the appearance of mist in the interior of the peninsula. These conditions usually occur when the Iberian Peninsula is under anticyclonic conditions in the

winter months [25], as reflected in Figure 6b, which shows the average value of the pressure field at sea level (SLP) for the month of January, 1799 [26].

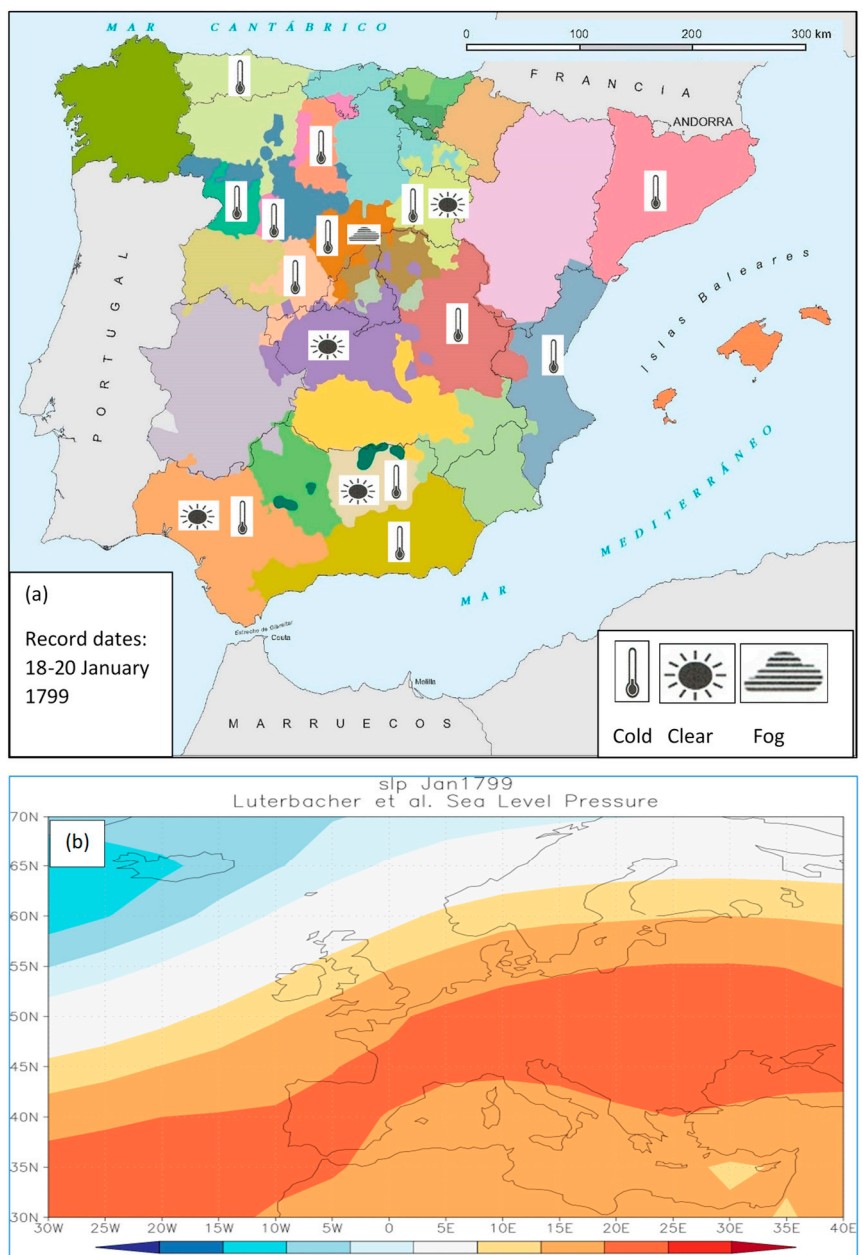

**Figure 6.** (**a**) Weather information corresponding to 11–19 January 1799 (reports dated from 18 to 20 January 1799, issues 6–11, 1799, CMEI). (**b**) SLP field in January 1799.

Figure 7a shows an example of a cold and wet spring. In this case, the records were dated between April 12th and 14th, 1793, indicating the predominant conditions between April 5th and 12th of that year (Table S2 in the Supplementary Material). The records for April 13th in Sevilla and Málaga indicate abundant rains. Heavy rains and snowfall were widespread in the IP, except for two eastern provinces, Murcia and Cuenca. This map can be interpreted in terms of the predominance of cyclonic conditions with north-westerly flows of Atlantic origin, a typical spring circulation [27], as can be seen in the reconstruction of the average SLP field for April 1793 (Figure 7b).

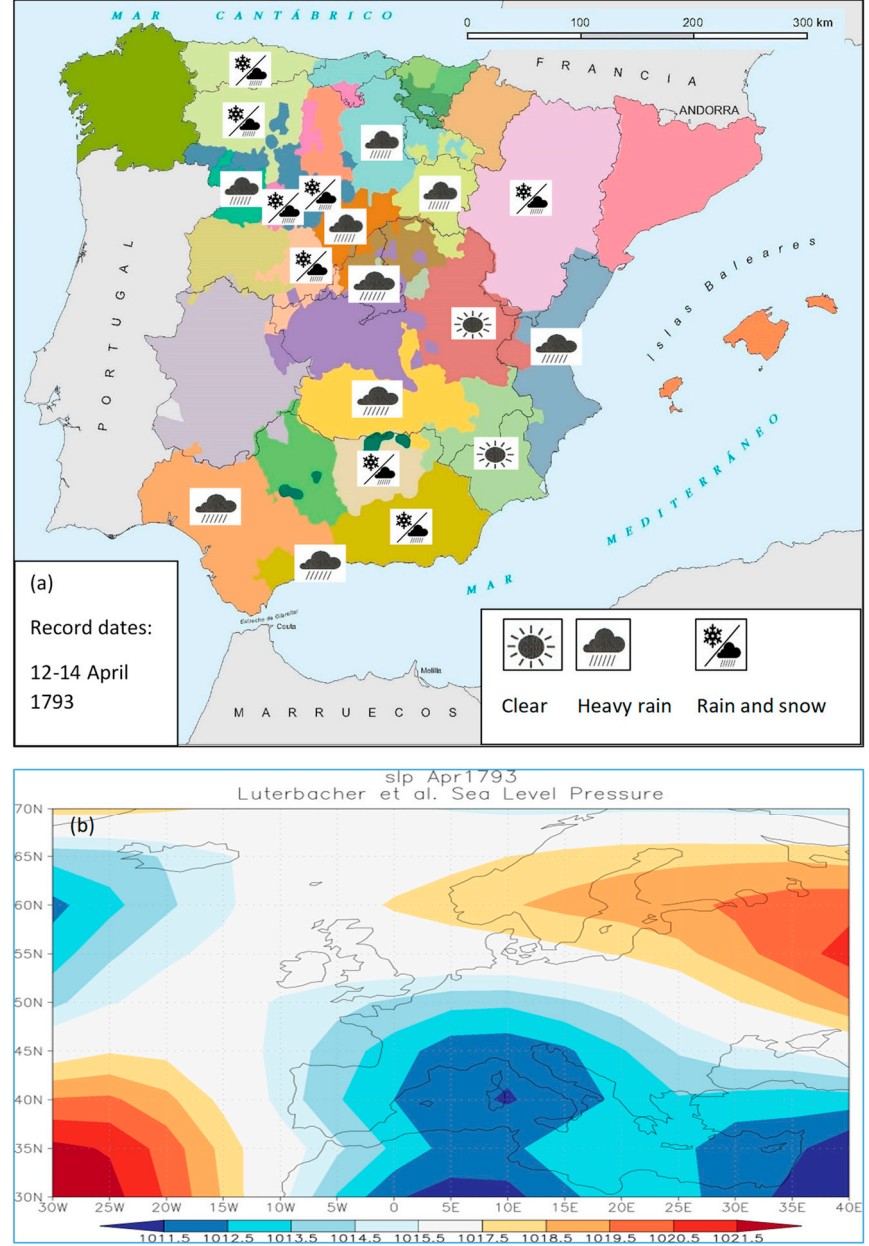

**Figure 7.** (**a**) Weather information corresponding to 5–13 April 1793 (reports dated from 12 to 14 April 1793, issuexs 8–11, 1793, CMEI). (**b**) SLP field in April 1793.

Figure 8a shows an example of warm, dry summers. These are records dated between July 15th and 17th, 1796, thus indicating the conditions for the period between 8th and 16th July of that year (Table S3 of the Supplementary Material). It highlights the information on easterly winds in the province of Burgos and the mild temperatures in Valencia, probably resulting from predominant anticyclonic conditions, which would cause easterly flows across the IP of Mediterranean origin, helping to smooth the temperatures on the Mediterranean front [28]. Figure 8b shows the reconstruction of the average SLP field in July 1796.

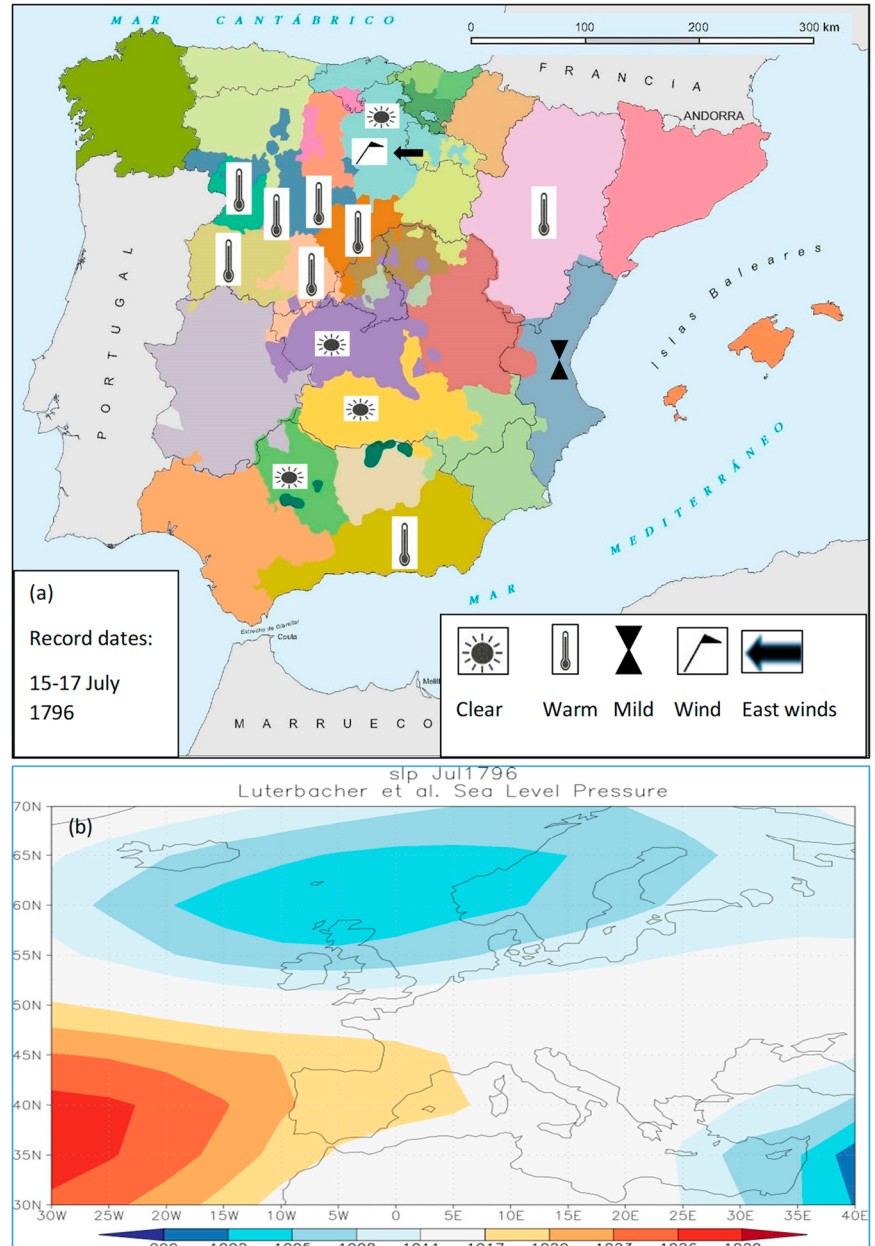

**Figure 8.** (**a**) Weather information corresponding to 8–16 July 1796 (reports dated from 15 to 17 July 1796, issues 57–60, 1796, CMEI). (**b**) SLP field in July 1796.

Figure 9a shows an example of autumn conditions. This corresponds to records dated between 21st and 23rd November 1794 (Table S4, Supplementary Material), thus indicating the prevailing conditions between 14th and 22nd November of that year. There was rainfall all across the western half of the IP, which was especially intense in the southwestern provinces, whereas the weather in the east was dry, in Aragon and Murcia. There was cold weather to the north in Galicia and Catalonia. These conditions seem to reflect the advection of moist air masses of Atlantic origin [27], as seen in the reconstruction of the average SLP field corresponding to November of that year (Figure 9b), reflecting the predominance of a zonal circulation over the IP. Situations of this type would explain the differences found along the west–east axis at this time of the year.

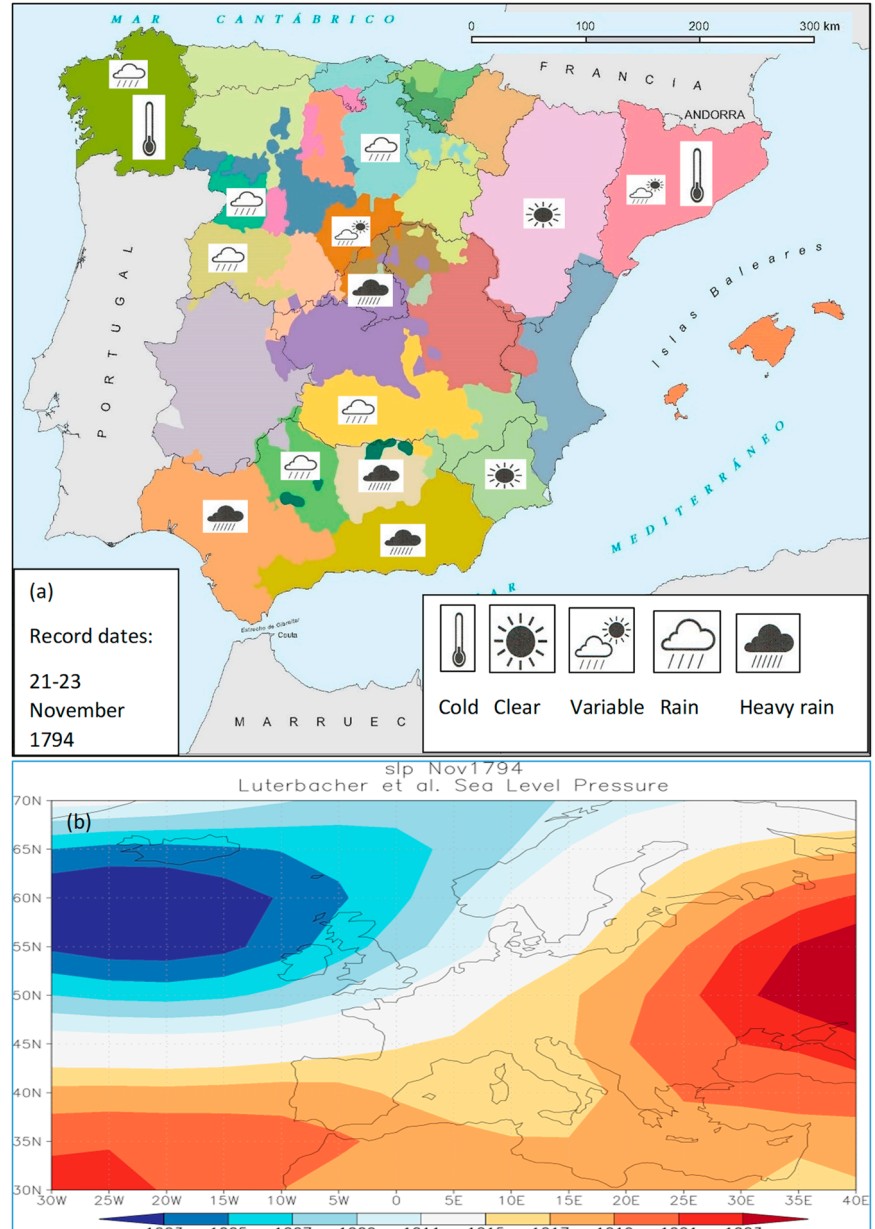

**Figure 9.** (**a**) Weather information corresponding to 14–22 November 1794 (reports dated from 21 to 23 November 1793, issues 94–99, 1794, CMEI). (**b**) SLP field November 1794.

## 4. Discussion

In the literature, we can find certain references to the climatic conditions in Spain during the study period. For example, Fernández-Fernández et al. [21] analysed the private correspondence between the major domo and the landowner in Zafra, Extremadura (near Sevilla and Córdoba) from 1750 to 1840. These authors found a dry period from 1796 to 1799 and a wet period between 1799 and 1807 [29], coinciding with the evolution of our $I_r$ index in the provinces of Sevilla and Córdoba (Figure 4). As for temperatures, these authors highlight the seasonal dependence of the thermal conditions [30] with high percentages of cold weeks in the winter of 1807 and the springs of 1799 and 1807, and of warm weeks in the winter of 1794, the autumn of 1797 and the summer of 1800, which coincide with our results for Sevilla and Córdoba (see Figure 5, and Figures S1 and S2, Supplementary Material). Regarding the driest conditions in Murcia, Alberola Romá [31] highlights the numerous pro-pluvia rogations performed between 1800 and 1807 in the south east of the IP.

Figure 10 shows the results comparing the $I_t$ index and the EMOSSv2 instrumental data in three cases: Granada (June 1796–September 1797; Gr1796–1797 in EMOSSv2), Cádiz (January 1799–December 1800; Ca1799–1800 in EMOSSv2), and Sevilla (October 1803–December 1806; Se1803–1806 in EMOSSv2; Cádiz and Sevilla belong to the same province in the CMEI). In all cases, the temperature was measured at noon. The monthly average of daily midday temperatures can be considered as a good approximation of the monthly average for the daily maximum temperatures [32]. In all three cases, the correlation coefficient between the monthly $I_t$ index and the average monthly temperature is significant at the 95% confidence level, despite the limited amount of data. If we accept the calibration shown in Figure 10, applying the regression equation to the average $I_t$ index value for the whole period allows us to estimate the average value of the daily maximum temperatures during the period studied. The results are 20 ± 2 °C in Granada, 18 ± 1 °C in Cádiz and 21 ± 1 °C in Sevilla. We can compare these values with the annual mean values corresponding to a modern reference period, such as the period from 1961–1990 [33]: 22.5 ± 0.1 °C in Granada, 21.3 ± 0.1 °C in Cádiz and 24.7 ± 0.1 °C in Sevilla. These results indicate that the study period was colder, in the order of 2 to 3 °C, than the reference period.

As for precipitation, there is little quantitative information on monthly accumulated rainfall in EMOSSv2. In the EMD, the main rain-related variable is the number of rainy days (RD). The RD has a strong correlation with the cumulative precipitation total but comparing it to the $i_r$ index is difficult because this is an average monthly index, relating to intensity rather than the amount of rain. An added problem is that the CMEI data are spatial averages, whereas the data for the number of rainy days in EMOSSv2 are local. While we can expect similar thermal behaviour between nearby locations, precipitation is a discrete, discontinuous variable with greater spatial variability, especially in the case of convective rain. However, a first approximation was made in [10], showing that the correlation coefficient between $I_r$ and RD in Cádiz during 1799–1800 (Ca1799–1800 in EMOSSv2) was 0.64, statistically significant at the 95% confidence level. Therefore, the treatment of the precipitation data from the CMEI needs further research and will be the subject of future work.

In any case, these results should be interpreted with caution. The total period 1792–1808 is short, and in addition to the gaps resulting from lost copies of the periodical, we must consider that, in certain years, the number of records is insufficient to get a conclusive result. Moreover, vague and imprecise terms were not indexed. In some cases, it is possible to determine their meaning by comparing them to the contemporary EMD series from EMOSSv2. Such a comparison is only possible during the short periods of time when both data series overlap. Thus, for example, the concept of "variable weather" in Murcia, according to the CMEI, might be associated in 14 cases with the appearance of drizzle and in four cases with variable winds, clouds and clear skies, according to the data from the "*Diario de Cartagena*" periodical corresponding to the year 1807 (Car1807 in EMOSSv2). This ambiguity ($i_r = 1$ in the first case, $i_r = 0$ in the second) means we cannot be sure of the character of the records from other years because we cannot compare them. Since the EMOSSv2 series were used for contrasting the CMEI data, these terms were not indexed to avoid circularity problems.

However, as we have seen, some insight can be gleaned about the general climatic conditions during the study period. This result suggests the existence of temperature and rain gradients, with an increase (decrease) in temperature (rain) from west to east. In an earlier work [34], these differences along the west–east axis were explained as being the result of the declining (growing) influence of Atlantic (Mediterranean) perturbations.

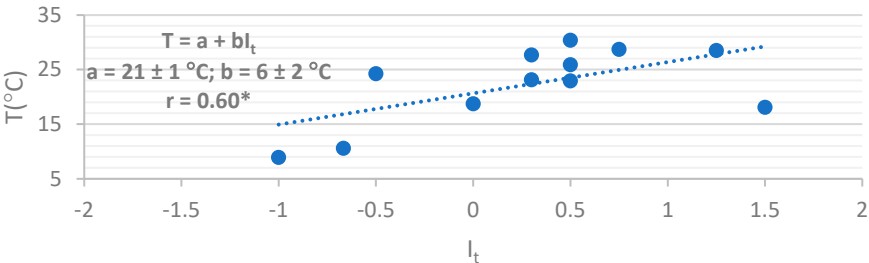

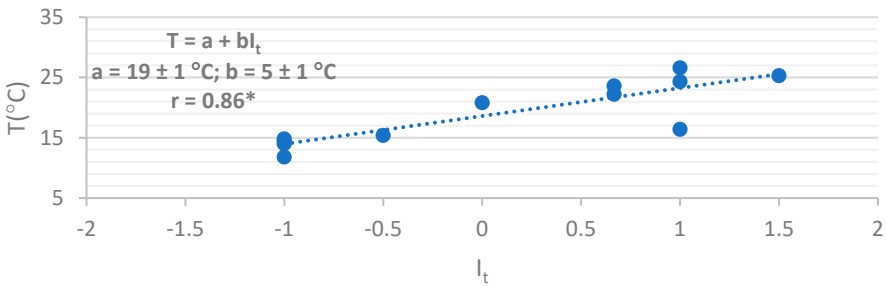

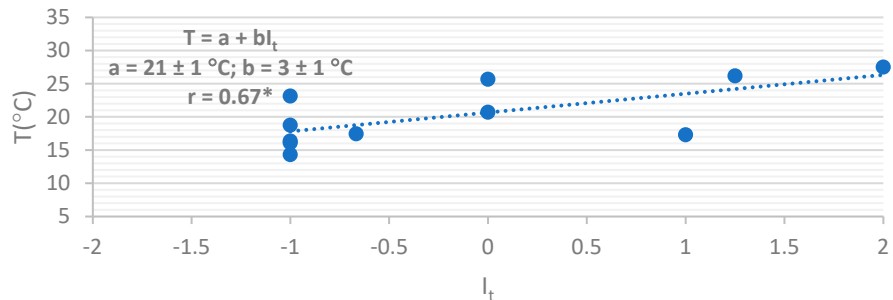

**Figure 10.** Comparison between monthly indices $I_t$ from CMEI and monthly mean temperatures at midday from EMD (EMOSSv2) in (**a**) Granada, June 1796 to September 1797, (**b**) Cádiz, January 1799 to December 1800, and (**c**) Seville, October 1803 to December 1806. Linear regression equations and correlation coefficients are included (* = significant at the 95% confidence level).

The relationship between winter rainfall in the IP and the North Atlantic Oscillation (NAO) is well known [35]: the positive phase of the NAO is related to an intensification of high pressures to the west of the IP, producing dry conditions and droughts in the area [36], while the negative phase diverts Atlantic cyclones to the south, so that they invade the IP, generating heavy rains, which can sometimes cause river flooding [37]. Although our analysis period is brief, we can compare our results with the independent NAO reconstruction by Luterbacher et al. [38,39]: the dry winters of 1793, 1796, 1798, and 1801 correspond to positive index values (+0.60, +0.83, +0.51 and +0.12, respectively), while wet winters, such as those of 1794, 1795, and 1800, correspond to negative NAO index values (−0.23, −0.84 and −1.27, respectively), reflecting this negative relationship in our data.

The results suggest a positive (negative) correlation between temperatures and precipitation in winter (the rest of the seasons). This implies a predominant pattern of cold–dry (or alternatively warm–wet) winters, while cold–wet (warm–dry) conditions appear in the other seasons. This pattern

has already been found in a previous study on the covariability of temperatures and precipitation in the IP during the period 1951–2016 [40], as well as in an analysis of climate variability in Europe since 1766 [41]. Thus, our results suggest the predominance of cold–dry conditions in winter, cold–wet conditions in spring and warm–dry conditions in summer. The autumn results show less clearly defined behaviour, according to the analysis by Casty et al. [41]. They found a weaker relationship between temperatures and precipitation than in spring in the Mediterranean Basin.

The period 1792–1808 coincides with the central years of the Dalton Minimum of solar activity. In addition, frequent volcanic activity was reported during this period [42]. The resultant reduction in short-wave radiation would explain the cooling detected in the winter and spring. A possible consequence would be the predominance of the positive phase of the NAO in winter, with anomalous dry conditions over the IP [43], and the negative phase of the East Atlantic Pattern (EA) in spring, causing an increase in precipitation over the Mediterranean Basin [44]. The interaction between these climate variability patterns could explain the climate variability observed in the IP [45] although further research is needed before final conclusions can be established.

## 5. Conclusions

The main objective of this work was to present an initial analysis of the climate data recorded in the CMEI during the period 1792–1808 relating to the south of the IP. This short period is interesting because it includes the Dalton Minimal of solar activity. Hence, this data could shed light on IP climate fluctuations during a brief period dominated by natural radiative forcing. The analysis has focused on the southern IP, showing the potential usefulness of this data source for future research. In particular, it allows us to view the spatial distribution of weather phenomena on a weekly scale. In addition, it provides information on areas (Jaén, Córdoba) that have had no known data from this period up until now.

Generally speaking, the results are in good agreement with previous results on weather conditions in the southern IP during the Dalton Minimum: cold conditions and a variable rainfall regime, particularly dry in winter and wet in spring. However, there is much work to be done. The study needs to be extended to other provinces for which information exists, a catalogue of extreme phenomena needs to be prepared, considering their intensity and spatial extension, and the defined indices need to be calibrated and validated (especially the monthly rainfall index, $I_r$) to obtain a quantitative reconstruction of climatic conditions in the area. All these aspects will be the subject of future work.

**Supplementary Materials:** The following are available online at http://www.mdpi.com/2073-4433/11/3/296/s1, Figure S1. (a) Number of seasonal records in Seville, 1792–1808, and combined temperature-rainfall conditions for winter (b), spring (c), summer (d), and autumn (e). Years are indicated by the last two digits. Figures S2 to S5, as Figure S1 for Córdoba, Jaén, Granada, and Murcia, respectively. Table S1. Reports dated from 18 to 20 January 1799 (issues 6–11, 1799, Figure 6a). Table S2. Reports dated from 12 to 14 April 1793 (issues 8–11, vol. 2, 1793, Figure 7a). Table S3. Reports dated from 15 to 17 July 1796 (issues 57–60, 1796, Figure 8a). Table S4. Reports dated from 21 to 23 November 1794 (issues 94–99, 1794, Figure 9a).

**Funding:** This research received no external funding.

**Acknowledgments:** The author wishes to express his gratitude to the anonymous referees for their useful comments.

**Conflicts of Interest:** The author declares no conflict of interest.

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
