# Peer review of "Recovering Climate Data from Documentary Sources: A Study on the Climate in the South of Spain from 1792 to 1808"

_atmosphere, doi:10.3390/atmos11030296_

Round 1

Reviewer 1 Report

Enlarging the observational records on local, regional to global climate represents a worthwhile and important undertaking in climate sciences. Thus, the current paper provides valuable information on weekly climate records for selected provinces of Spain from 1792 to 1808. This is based on reports published in a national newspaper (Correo Mercantil de España y sus Indias), which have become recently available in digitized form.

After a slightly lengthy Introduction (should be shortened), the author provides the data sources and methods used in his study. The 3rd section presents the main results obtained, which are discussed in the following 4th section, where comparisons to other data and results are provided. The well-prepared supplemental material offers additional data and results. This is followed by a few conclusions and some suggestions for additional work.

The structure of the paper is well-conceived and provides for straightforward understanding of the material provided. The figures are mostly well prepared and informative. The paper is relatively well written, however, it will need a significant language check and quite a number of reformulations. I have marked a few places that need improvement in the first sections of the paper, but have not doe this throughout.

A major shortcoming of the paper –in my view- lies in the near-absence of a sufficient analysis/discussion on errors and uncertainties of the results presented. While the author states (p. 12, L. 242-243): The best proof for the reliability of these results is to check other evidences from independent data sources to see if they show similar characteristics, the “checks” performed remain largely qualitatively. I am not sure what exactly the comparisons between his results and Sea Level Pressure (SLP) fields as derived by Luterbacher et al. (2002; 2020) reveal. Is there good or moderate or no agreement between the data sets? The explanations provided on p. 11 (L. 231-237), while reasonable do not provide much of a “proof”. However, if this is all that can be done in such cases, the author should explain this explicitly (e.g., in the Methods section). In any case, a more substantial discussion on the overall “value” of such analyses, in light of its apparent reliability would improve the paper.

Another major (ethical) issue that I have with the paper is the fact that at least parts of it have been or are being published elsewhere, i.e., in Rodrigo, F. S. (2020) NEW DOCUMENTARY DATA ON THE CLIMATE IN SOUTHERN SPAIN FROM 1792 TO 1808, Geographical Research Letters, No. 46, http://doi.org/10.18172/cig.4290. For instance, the sentence cited above (p. 12, L. 242-243) is almost identical to the sentence in the mentioned paper: “The best proof for the reliability of this data source is to check other evidence from different data sources (qualitative and/or quantitative) to see if they show similar characteristics.” Or, what has been said on p. 15, L. 328-331, is very similar to the first two sentences in the Conclusion section of Rodrigo (2020). –Unfortunately, I could not check how similar the current paper is t the one published in 2019 (Rodrigo, F. S (2019) Early meteorological data in southern Spain during the Dalton Minimum , Int. J. Climatol., https://doi.org/10.1002/joc.6041), since I do not have access to this journal. However, based on the Abstract, there seem to be similarities as well.

Reviewer 2 Report

This manuscript describes efforts to quantify climate-related information from a documentary source relating to southern Spain in the period 1790-1810.  The documentary references provide useful calibration with and validation of the fragmentary instrument records for that time period in the region presented earlier by the author. As a result, I think the results are important and worthy of publication.

The basic research design and analyses are straightforward and are presented reasonably well, so I don't see any substantive issues that need attention. The paper does need careful editorial scrutiny, however, to address numerous problems with English grammar and usage. The Spanish-English translations of weather-related terminology in Table 1, for example, seem overly literal. Other than fixing these stylistic problems, however, I don't think the paper needs any further revision.

Round 2

Reviewer 1 Report

As mentioned before, enlarging the observational records on local, regional to global climate represents a worthwhile and important undertaking in climate sciences. Since the current paper does provide valuable information on (qualitative) weekly weather records for selected provinces of Spain from 1792 to 1808, it should therefore be published in atmosphere.

In general, the author has done a good and thorough job to improve the paper by following the suggestions of the reviewer in the revised version presented. However, instead of having shortened the Introduction (in the first version comprising lines 16-81, i.e., 65 lines), it is now (apparently longer), i.e. comprising lines 17-95, i.e., 78 lines. Have I misunderstood anything here?

However, more substantially, the author has provided satisfactory explanations as to the reliability and the analysis/discussion on errors and uncertainties of the results presented. It is clearly stated now in the paper that “…the study of its reliability (i.e., ensuring that what they describe actually occurred) is itself essentially qualitative”. The author therefore states appropriately in the Discussion section “…these results should be interpreted with caution…”. The remarks regarding the comparison with the Luterbacher et al. data (“…These reconstructions allow us to interpret in a plausible way the dynamic atmospheric conditions that gave rise to the events in our maps…”) are also well stated.

The explanations provided regarding the issue of repeating earlier publications, while understandable in general, still raises the issue to what extent a “…split it into two papers…” of material, genuinely comprising one study is appropriate? While I am well aware that this is frequently done, this does –in my humble view though- not justify such practices.

There are a few smaller issues to be noted (this is by no means comprehensive, but rather indicative):

  • p. 2, l. 65: “providing data information” should be changed to “providing information”;
  • p. 2, l. 68: “In in the present study…” should be changed to “In the present study…”;
  • p.7, l.242: the term “EMOSSv2” should by spelled out in the text (“Early Meteorological Observations in southern Spain”) and not only in the reference list;
  • Figure 5 is largely missing from the current manuscript;
  • The figure captions to parts (b) (SLP field) of Figures 6 to 9 are hardly readable; this should be corrected.

These (and possibly other) minor insufficiencies should be corrected, before the paper may be published.
